# Embracing Consistency: A One-Stage Approach for Spatio-Temporal Video Grounding

**Yang Jin[1], Yongzhi Li[2], Zehuan Yuan[2], Yadong Mu[1,3]***

[1]Peking University    [2]ByteDance    [3]Peng Cheng Laboratory

`jiny@stu.pku.edu.cn, liyongzhi.ailab@bytedance.com,`
`yuanzehuan@bytedance.com, myd@pku.edu.cn`

## Abstract

Spatio-Temporal video grounding (STVG) focuses on retrieving the spatio-temporal tube of a specific object depicted by a free-form textual expression. Existing approaches mainly treat this complicated task as a parallel frame-grounding problem and thus suffer from two types of inconsistency drawbacks: *feature alignment inconsistency* and *prediction inconsistency*. In this paper, we present an end-to-end one-stage framework, termed Spatio-Temporal Consistency-Aware Transformer (STCAT), to alleviate these issues. Specially, we introduce a novel multi-modal template as the global objective to address this task, which explicitly constricts the grounding region and associates the predictions among all video frames. Moreover, to generate the above template under sufficient video-textual perception, an encoder-decoder architecture is proposed for effective global context modeling. Thanks to these critical designs, STCAT enjoys more consistent cross-modal feature alignment and tube prediction without reliance on any pretrained object detectors. Extensive experiments show that our method outperforms previous state-of-the-arts with clear margins on two challenging video benchmarks (VidSTG and HC-STVG), illustrating the superiority of the proposed framework to better understanding the association between vision and natural language. Code is publicly available at https://github.com/jy0205/STCAT.

## 1 Introduction

Visual grounding is a prominent and fundamental task in the multi-modal understanding field. It aims to localize a region from the visual content specified by a given natural language query. As a bridge to connect vision and language, this topic has drawn increasing research attention over the past few years [13, 11, 45, 17, 16]. Early works mostly focused on static images and have achieved remarkable progress, while visual grounding in videos has not been adequately explored yet. Recently, spatio-temporal video grounding (STVG) was introduced in [48], which is a compound task, requiring both spatial and temporal localization. Formally, given an untrimmed video and a textual description of an object, this task aims at producing a spatio-temporal tube (*i.e.*, a sequence of bounding boxes) for the queried target object (See Figure 1). Compared with previous grounding tasks in images, STVG is substantially more challenging as it delivers high demands for distinguishing subtle spatio-temporal status of instance in videos based on the query sentence. Thus, how to effectively align the textual semantics and time-varying visual appearance is especially critical for settling this task.

The majority of current approaches [38, 34, 4, 48, 47] streamline the spatio-temporal video grounding as a two-stage pipeline, where the object proposals or tubelets are firstly generated by a pre-trained object detector (*e.g.*, Faster RCNN [25]) and then all the candidates are ranked according to the similarity with the query. However, this simple design leads to an inevitable deficiency: the grounding

---

*Corresponding Author.

**Query Sentence:**

*An adult in blue grabs a ball on the basketball court*

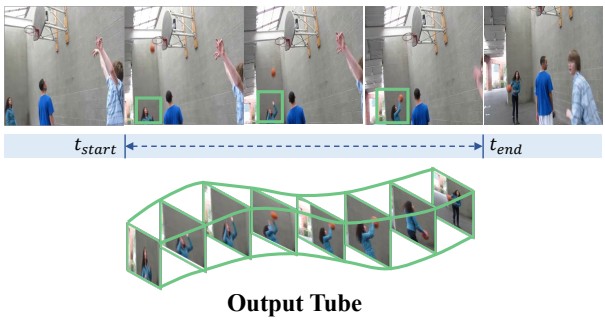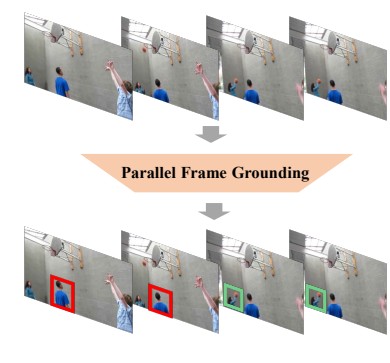

$t_{start}$ ⟵ - - - - - - - - - - - - - - - - - - - - ⟶ $t_{end}$

**Output Tube**

**Parallel Frame Grounding**

**Inconsistent Prediction**

Figure 1: An illustration of spatio-temporal video grounding task (left) and the inconsistency prediction in previous approaches (right). This task is particularly challenging and requires exquisite reasoning about linguistic semantics and the global video context. Existing methods simply treat STVG as a parallel image grounding problem, leading to inconsistent predictions (highlighted in red).

performance is heavily limited by the quality of proposals, especially when a huge domain gap exists between the pre-trained object categories and free-form text. Lately, some approaches [29, 33] try to address these issues by leveraging a one-stage framework without any off-the-shelf object detectors. They directly produce the bounding box at each frame as well as predict the starting and ending boundary based on the fused cross-modal features.

Though promising results have been achieved, most of these one-stage approaches have the following drawbacks that need to be concerned. (1) **Feature alignment inconsistency**. Global context modeling is of great significance for the STVG scenario. For example, given the query of "An adult in blue grabs a ball on the basketball court" as in Figure 1, the target "adult" cannot be determined without a perception of the whole video content, since there are two adults in blue but only one of them eventually grabs the ball. Accordingly, the cross-modal feature alignment should be performed between the whole video and the querying sentence. However, existing methods [48, 29, 33] only attend the cross-modal fusion with local short-term video context. Even in the very recent method [40], the alignment is still considered at frame-level. (2) **Prediction inconsistency**. Previous approaches simply treat STVG as a parallel frame-level grounding problem. As shown in Figure 1, they separately generate a bounding box with respect to the query on every single frame, yet neglect the consistency among frames. It is worth noting that the goal of STVG task is to localize a single target instance specified by the textual description. Although the appearance of the target instance across all video frames may change due to camera motion, scene dynamics, etc., they should have identical semantics (*e.g.*, the guy who grabs the ball). Therefore, conducting grounding independently on each frame regardless of semantic consistency over time will cause the prediction mismatch.

In this paper, we present an effective one-stage transformer-fashion encoder-decoder based [35] framework, dubbed as Spatio-Temporal Consistency-Aware Transformer (STCAT) to cope with the aforementioned issues in the STVG task. Building upon the prevalent detection network DETR [2], STCAT introduces a novel video-level multi-modal template to modulate the grounding procedure. This template serves as a global target that guides and correlates the predictions among each frame to maintain temporal consistency. Specially, it is formulated as a learnable query as in [2], which is comprised of a content term shared by all frames to encode the semantics and a position term characterized by per frame. Such design allows the final grounding results to be determined based on the specific frame content while simultaneously considering consistency. Moreover, in order to adaptively obtain the above-mentioned query by considering the interactions between visual and textual information, we propose a cross-modal spatio-temporal encoder to fuse the global video context and textual description. The query in STCAT is then rendered based on the encoded features and fed into a decoder to aggregate the multi-modal representation. During the decoding procedure, the query belonging to each frame is dynamically updated layer by layer. Finally, the grounded spatio-temporal tubelet is produced by different prediction heads.

The technical contributions of this work can be summarized as follows. (1) We propose a novel multi-modal template mechanism to mitigate the prediction inconsistency issue in the STVG task. In doing so, the grounding among all video frames can be correlated to yield a more precise spatio-temporal

prediction. (2) We design a transformer-based architecture, named as STCAT, to fully exploit the global video context and thus perform a better cross-modal alignment. Coupling with the proposed template mechanism, our STCAT can directly ground the spatio-temporal tubes without reliance on any pre-trained object detectors. (3) Comprehensive experiments conducted on two challenging video benchmarks (VidSTG [48] and HC-STVG [33]) further demonstrate that our method obtained new state-of-the-art performance compared to other approaches.

## 2  Related Work

**Visual Grounding in Images / Videos.**    Visual grounding is an essential multi-modal task that aims to localize the object of interest in an image/video based on a text description. In the image grounding scenario, most existing methods [18, 44, 20, 36, 41] firstly utilize a pre-trained object detector to get some object proposals. While some recent works like [16, 43, 23, 42] propose one-stage frameworks that alleviate the reliance on pre-trained detectors. For example, in [16] Liao *et al.* proposed to extract cross-modal representations and localize the target objects by an anchor-free object detection method. Yang *et al.* tried to generate text-conditional visual features by sub-queries in [42], which further promotes the one-stage method.

The video grounding task can be categorized into temporal grounding and spatio-temporal grounding. The former requires to localizes a temporal clip in the video by referring sentence. While the spatio-temporal video grounding lies at the intersection of spatial and temporal localization. Most previous approaches [4, 33, 38, 48] also rely on pre-extracted tube or object proposals. Recently, STVGBert in [29] propose a one-stage approach that extend the VilBERT [22] to settle this task. However, all these methods suffer from the drawbacks mentioned in Section 1. More recently, a concurrent work in [40] also proposed a one-stage framework that utilizes DETR [2] architecture. However, they only consider the multi-modal interaction at the frame level and still lack an effective design for settling prediction inconsistency in STVG, thus achieving inferior performance to ours.

**Vision-language modeling.**    Vision-language modeling tasks such as visual question answering, image captioning, and image-text retrieval attract lots of researchers to explore recently. Due to the simplicity and success of Transformers in the natural language processing field, many works [32, 14, 22, 30] borrow this architecture to align the context information between images and sentences. Others like [31, 50, 6, 9, 15, 39] further extend transformers into video-text tasks. But most of them rely either on pre-extracted object features, or spatially pooled features, that ignored detailed spatial information within each frame and are not capable to handle the challenging STVG task.

**Transformer based detection.**    Object detection is a traditional task in the computer vision field, early methods [25, 24] usually rely on CNN encoders and region proposal or regression modules to tackle the various objects in images. As Carion *et al.* [2] first introduce Transformers into this task, a lot of transformer-based detection methods [51, 1, 49] have sprung up recently. Most of them follow the encoder-decoder paradigm to transform the visual features into precision object bounding boxes, we borrow this effective architecture to tackle the STVG task in this paper.

## 3  The Proposed Approach

In this section, we briefly present the formal definition of the STVG task and an overview of our proposed framework in Section 3.1. Then we elaborate on the main components of STCAT, including the feature extractor (Section 3.2) and the consistency-aware transformer (Section 3.3). Finally, the training and inference details are introduced in Section 3.4.

### 3.1  Overview

Given an untrimmed video $\mathbf{V} = \{v_t\}_{t=1}^{T}$ consisting of $T$ consecutive frames and a query textual description $\mathbf{S} = \{s_n\}_{n=1}^{N}$ depicting a target object existing in $\mathbf{V}$. The goal of the STVG task is to localize a spatio-temporal tube $\mathbf{B} = \{b_t\}_{t=t_s}^{t_e}$ corresponding to the semantics of given textual query, where $b_t$ represents a bounding box in the $t$-th frame, $t_s$ and $t_e$ specify the starting and ending boundary of the retrieved object tube respectively. STVG is an extremely challenging task, which requires not only the spatial interaction with the textual modality for detecting a bounding box at the

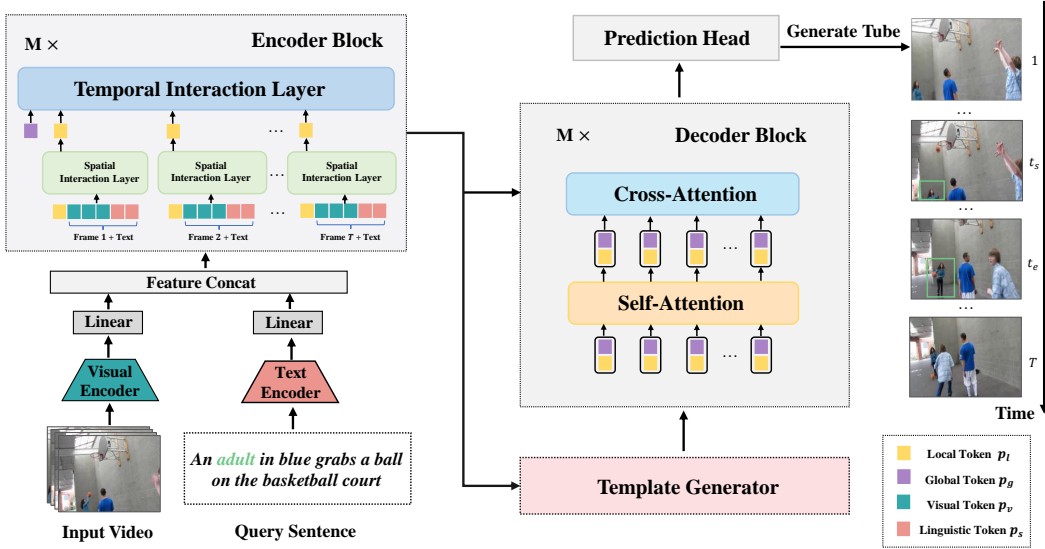

Figure 2: The Architecture of the proposed Spatio-Temporal Consistency-Aware Transformer (STCAT). Given an input video and query sentence pair, it firstly leverages a visual and a linguistic encoder to extract features for each modality. The extracted features are fed into a encoder to perform cross-model interaction. Then a generator yields the multi-modal templates which is responsible for guiding the decoder. The retrieved object tube is finally generated based on the decoded features via a prediction head.

frame level, but also the long-range temporal relation modeling at the video level for determining the start and end timestamps of query-related video segments.

To mitigate the aforementioned feature alignment and prediction inconsistency drawbacks, an effective framework named STCAT is developed in this paper. As illustrated in Figure 2, the proposed model firstly utilizes two feature extractors to obtain both visual and textual features from the video frames and querying sentence, respectively. It then models the video-text interactions through a well-designed spatio-temporal cross-modal encoder, which introduces a global learnable token for the whole video to encode target object semantics, and a local one for each individual frame to represent the frame-specific appearance. These tokens are further leveraged to produce a template for the target object by a template generator. Finally, the yielded template is treated as a query (Like the one in DETR [2]) and fed into a decoder to aggregate features and predict the retrieved spatio-temporal tube.

### 3.2 Feature Extractor

**Visual Encoder.** We start by adopting a vision backbone (*e.g.*, ResNet [10]) to extract the visual features for each frame $v_t$ individually in video $\mathbf{V} = \{v_t\}_{t=1}^T$. The obtained 2D feature map of per frame is then flattened, bringing a visual feature sequence $\mathcal{F}_v = \{f_t^v\}_{t=1}^T$. Each $f_t^v \in \mathcal{R}^{N_v \times C_v}$ serves as a compact representation for $v_t$, where $N_v = H \times W$ and $C_v$ is the visual feature channel.

**Linguistic Encoder.** For the querying sentence $\mathbf{S} = \{s_n\}_{n=1}^{N_S}$ with $N$ words, we leverage a pre-trained linguistic embedding encoder, (*e.g.*,BERT model [5]), to encode the linguistic representation. The textual features of input sentence query is denoted as $\mathcal{F}_s = \{f_i^s\}_{i=1}^N$, where $f_i^s \in \mathcal{R}^{C_s}$ and $C_s$ is textual feature channel.

### 3.3 Consistency-Aware Transformer

The detailed spatio-temporal video grounding procedure is executed by leveraging four crucial components: a cross-modal transformer encoder, a template generator, a query-guided transformer decoder, and two parallel prediction heads.

#### 3.3.1 Multi-modal Encoding

The proposed encoder is obliged to fully exploit the cross-modal interaction between the video and text, correlating their corresponding semantics in a fine-grained manner. Existing one-stage

video grounding approaches only restrict this interaction either in individual frames [40] or short video snippets [29]. This design leads to an inconsistent cross-modal feature alignment as the query sentence depicts a long-term evolutional event in videos.

To this end, the developed encoder aims at performing a more consistent feature alignment between two modalities. Specially, given the input features $\mathcal{F}_v$ and $\mathcal{F}_s$, a projection layer is first applied to embed them into the same channel dimension $C$. We denote the projected visual embedding as $p_v = \{p_{v_t}\}_{t=1}^T$, where $p_{v_t} \in \mathcal{R}^{N_v \times C}$ and linguistic embedding as $p_s \in \mathcal{R}^{N_s \times C}$. Besides, we introduce a learnable token denoted as $p_g \in \mathcal{R}^C$ to integrate the global video context during encoding, and $T$ analogous ones dubbed as $p_l = \{p_l^t \in \mathcal{R}^C\}_{t=1}^T$ to attend the local context within individual frame. The proposed encoder reads all above-described embedding tokens as its input and has $M$ stacked encoder blocks. Each block consists of a spatial and a temporal interaction layer. Both adopt the transformer-style encoder structure [35].

**Spatial Interaction Layer.** The goal of this layer is to conduct the intra- and inter-modality relation modeling spatially for each local frame. In detail, the input $x_t$ for each interaction layer on the $t$-th frame can be formulated as:

$$x_t = [p_l^t, \underbrace{p_{v_t}^1, p_{v_t}^2, ..., p_{v_t}^{N_v}}_{\text{visual tokens } p_{v_t}}, \underbrace{p_s^1, p_s^2, ..., p_s^{N_s}}_{\text{linguistic tokens } p_s}]. \tag{1}$$

The joint input sequence $x_t$ is then fed into one spatial interaction layer to yield the contextualized visual-text representations for each frame respectively. It is worth noting that the state of frame-specific token $p_l^t$ is enriched by both visual and linguistic context within frame $v_t$.

**Temporal Interaction Layer.** The aforementioned spatial interaction layer only attends the local context information at frame level but lacks the modeling of global context across the whole video. To address this, our temporal interaction layer further models the interactions temporally between local frames. Similarly, the input $x$ belonging to this layer is fed into another transformer encoder layer, which can be formulated as:

$$x_g = [p_g, \underbrace{p_l^1, p_l^2, ..., p_l^T}_{\text{frame tokens } p_l}]. \tag{2}$$

To retain the temporal relative information, we also add a positional encoding to $x_g$. Through this temporal layer, the local frame tokens $p_l = \{p_l^t\}_{t=1}^T$ attend the whole video content and the global token also aggregates the global video-text context.

After the above encoding procedure, we harvest the contextualized multi-modal features $F_{vl} \in \mathcal{R}^{T \times (N_v + N_s) \times C}$, the global embedding $p_g$ for the whole video and the local embedding $p_l = \{p_l^t\}_{t=1}^T$ with respect to all $T$ video frames.

### 3.3.2 Template Generation

Unlike previous works that ground each frame separately without considering consistency, we design a template-based mechanism to correlate and restrict the predictions across all video frames. Concretely, the developed template is comprised of a content term and a position term. The content term $q_c$ is shared by all frames to hold identical semantics depicted by the query sentence, while the position term $q_p = \{q_p^t\}_{i=1}^T$ is characterized by per frame according to its appearance. Both of them are yielded by a template generator based on the encoded local and global tokens.

Figure 3: The illustration of procedure for yielding content term $q_c$ and position term $q_p$ in the template generator.

As illustrated in Figure 3, the content term $q_c \in \mathcal{R}^C$ is generated by jointly considering the global visual-linguistic context:

$$q_c = W_c p_g + b_c, \tag{3}$$

where both $W_c$ and $b_c$ are learnable parameters. As for the position term $q_p = \{q_p^t\}_{i=1}^T$, where each $q_p^t$ is a 4-dimensional vector dubbed as $(x_t, y_t, w_t, h_t)$, which serves as a reference anchor to restrict the grounding region on each individual frame $v_t$. Based on considerations of both global semantics consistency and local characteristic specificity are critical

for $\boldsymbol{q_p}$, we generate it via depending both local frame embedding $p_l$ and global embedding $p_g$ (See Figure 3). In detail, inspired by [46, 42], we firstly leverage $p_g$ to modulate the local embedding $p_l = \{p_l^t\}_{t=1}^T$. It is fulfilled by two linear layers to generate two modulation vectors $\gamma^c \in \mathbf{R}^C$ and $\beta^c \in \mathbf{R}^C$ based on $p_g$:

$$\gamma^c = \tanh\left(W_\gamma p_g + b_\gamma\right), \quad \beta^c = \tanh\left(W_\beta p_g + b_\beta\right), \tag{4}$$

where $\boldsymbol{W_\gamma}$, $\boldsymbol{b_\gamma}$, $\boldsymbol{W_\beta}$ and $\boldsymbol{b_\beta}$ are learnable parameters. Finally, the position term $\boldsymbol{q_p^t}$ for each specific frame $v_t$ is obtained by modulating $p_l^t$:

$$\boldsymbol{q_p^t} = \mathrm{Sigmoid}\left(f_p\left(\gamma^c \odot p_l^t + \beta^c\right)\right), \tag{5}$$

where $f_p$ is a learnable mapping layer: $\mathcal{R}^C \to \mathcal{R}^4$.

### 3.3.3 Query-Guided Decoding

The objective of our decoder is to convert the multi-modal representations $F_{vl}$ obtained from the encoder to the target object tube based on the generated template. Inspired by recent query-based detectors [37, 19, 8], we formulate the above template $(\boldsymbol{q_c}, \boldsymbol{q_p})$ as object queries to guide the overall decoding procedure, where $\boldsymbol{q_c}$ serves as content queries to probe target object patterns in encoded embedding and $\boldsymbol{q_p}$ acts as positional queries to restrict the possible attending regions [19].

In detail, we denote $\{Q_t\}_{t=1}^T$ as the object query for each individual frame $v_t$. The initial query for feeding the decoder is formulated by $Q_t = [C_t; P_t]$ and we have:

$$C_t = \boldsymbol{q_c}, \quad P_t = \mathrm{Linear}(\mathrm{PE}(\boldsymbol{q_p^t})), \tag{6}$$

where PE means the sinusoidal position encoding to $\boldsymbol{q_p^t} = (x_t, y_t, w_t, h_t)$. For disentangling the spatial and temporal feature aggregation, we introduce a dual-decoder to handle the prediction of bounding-box and temporal boundary separately. Both adopt the same standard architecture in line with DETR [2]. Each decoder block includes a self-attention module for message passing across all video frames, and a cross-attention module for feature probing within the corresponding frame $v_t$. The content query $C_t$ is thus enriched by stacking $M$ decoder layers, resulting in the final contextualized representation used for generating object tubes. Following the previous practice [51, 37], the position query $P_t$ in bounding-box decoder is updated after each layer via predicting a relative offset with respect to $\boldsymbol{q_p^t}$ by a shared regression head described below. More concrete exposition for the query-guided decoding procedure is presented in supplementary material.

**Prediction Head.** After decoding, the refined content query $C_t$ is utilized for simultaneously predicting spatial and temporal localization. Specially, a regression head implemented as a 3-layer MLP is adopted to perform bounding box coordinates prediction. The output of this prediction head is a 4-dimensional coordinate offset $(\Delta x_t, \Delta y_t, \Delta w_t, \Delta h_t)$ and the final predicted box $\hat{b}_t \in [0, 1]^4$ in frame $v_t$ is obtained by:

$$\hat{b}_t = (x_t + \Delta x_t, y_t + \Delta y_t, w_t + \Delta w_t, h_t + \Delta h_t). \tag{7}$$

The temporal boundary $[\hat{t}_s, \hat{t}_e]$ is generated by predicting the start and end probabilities of each frame through a similar MLP prediction head.

### 3.4 Training and Inference

**Training.** At the training stage, we send a batch of video-sentence pairs to the proposed model. Each pair has a ground-truth bounding box sequence $\mathbf{B} = \{b_t\}_{t=t_s}^{t_e}$ and the corresponding start and end timestamps. For spatial localization, we involve the box prediction loss $\mathcal{L}_{bbox}$ as follows:

$$\mathcal{L}_{bbox} = \lambda_{L_1}\mathcal{L}_{L_1}(\hat{B}, B) + \lambda_{\mathrm{giou}}\mathcal{L}_{giou}(\hat{B}, B), \tag{8}$$

where $\mathcal{L}_{L_1}$ and $\mathcal{L}_{\mathrm{giou}}$ are the smooth $L_1$ loss and generalized IoU loss [26] on the bounding boxes respectively. Note that $\mathcal{L}_{bbox}$ only considers predictions in $[t_s, t_e]$. As for temporal localization, we follow [27, 29, 40] to generate two 1-dimensional gaussian heatmaps $\pi_s, \pi_e \in \mathcal{R}^T$ for starting and ending positions. And the temporal prediction loss is termed as $\mathcal{L}_{\mathrm{temp}} = \mathcal{L}_s(\hat{\pi}_s, \pi_s) + \mathcal{L}_e(\hat{\pi}_e, \pi_e)$, where $\mathcal{L}_s$ and $\mathcal{L}_e$ are the KL divergence between target and predicted distributions. To encourage a more accurate temporal prediction, we also add an auxiliary head to predict whether a frame belongs to the ground-truth segment. It is supervised by a binary cross-entropy loss term $\mathcal{L}_{\mathrm{seg}}$. The total training loss is defined as:

$$\mathcal{L} = \mathcal{L}_{bbox} + \lambda_{\mathrm{temp}}\mathcal{L}_{\mathrm{temp}} + \lambda_{\mathrm{seg}}\mathcal{L}_{\mathrm{seg}}. \tag{9}$$

**Inference.** During inference, our framework produces the bounding boxes and the starting and ending probabilities for all frames. The start and end times of output tube, $\hat{t}_s$ and $\hat{t}_e$, are determined by selecting segment with maximal joint starting and ending probability. Finally, the bounding boxes within $[\hat{t}_s, \hat{t}_e]$ form the retrieved object tube.

## 4 Experiments

### 4.1 Datasets and Metrics

**Datasets.** To evaluate the proposed method, we follow the previous work [29] and adopt two large video grounding benchmarks: **VidSTG** [48] and **HC-STVG** [33]. Both two datasets are annotated with spatio-temporal tubes corresponding to text queries. VidSTG contains in total 6,924 untrimmed videos, which are clipped into 44,808 video-triplet pairs with 99,943 sentences describing 80 types of objects. We follow the pioneer work [48] to split all videos into three parts, 5,563 for training, 618 for validation, and 743 for testing, which paired with 80,684, 8,956, and 10,303 distinct sentences respectively. HC-STVG contains 5,660 untrimmed videos in multi-person scenes, each annotated with one expression related to human attributes or actions. This dataset is divided into the training set and the testing set with 4,500 and 1,160 video-sentence pairs, respectively.

**Evaluation Metrics.** We follow [48] and select **m_vIoU**, **m_tIoU** and **vIoU@R** as the evaluation criteria. Note that $\text{vIoU} = \frac{1}{|S_u|} \sum_{t \in S_i} \text{IoU}(\hat{b}_t, b_t)$, where $S_i$ and $S_u$ are the intersection and union between the predicted tubes and ground-truth tubes, respectively. And $\text{IoU}(\hat{b}_t, b_t)$ describes the IoU score between the detected bounding box $\hat{b}_t$ and ground-truth bounding box $b_t$ at frame $t$. While the m_vIoU score is defined as the average vIoU score over all testing videos. As for vIoU@R, which is the ratio of samples whose vIoU > R in the testing subset. Besides, we also use m_tIoU (the mean of tIoU) to evaluate the temporal localization performance only, where $\text{tIoU} = \frac{|S_i|}{|S_u|}$.

### 4.2 Implementation Details

In line with the previous methods, we adopt the ResNet-101 [10] as the visual encoder and RoBERTa [21] as the linguistic encoder. The output from the $4$-th residual block of ResNet-101 is adopted as the extracted visual feature. For both encoder and decoder, the number of attention head is set to $8$ and the hidden dimension of feed-forward networks in the attention layer is $2,048$. Following [29, 40], part of the model parameters are initialized with the pre-trained weights provided in [12] and the whole framework is end-to-end optimized during training. Due to the appearance similarity between adjacent frames, the input video are uniformly down-sampled for computation efficiency. Besides, data augmentations include random resizing and random cropping are also applied to all training videos. The final object tube is obtained by linearly interpolating the predicted bounding-box in sampled frames. We empirically set the hyper-parameters in this paper with $M = 6, C = 256$ and the loss weights as $\lambda_{L_1} = 5, \lambda_{\text{giou}} = 3, \lambda_{\text{temp}} = 10$ and $\lambda_{\text{seg}} = 2$. The batch size and base learning rate are set to be $32$ and $1e^{-4}$, respectively. More implementation details are shown in supplementary material.

### 4.3 Performance Comparison

To fully demonstrate the superiority of proposed model, we compare with all previous spatio-temporal video grounding methods on VidSTG and HC-STVG datasets. Specially, there are three types of competitors: (a) **Factorized**: These approaches factorize the STVG task by performing spatial and temporal grounding separately. They firstly utilize temporal visual grounding techniques like TALL [7] and L-Net [3] to predict the temporal boundary of the target object and then perform spatial visual grounding with methods like GroundeR [28], STPR [38] and WSSTG [4] to generate the bounding boxes. (b) **Two-Stage**. The two-stage approaches like STGRN [48], STGVT [33] and OMRN [47] accomplish spatio-temporal grounding via firstly generating box proposals in each frame by means of a pre-trained object detector and then selecting the best matches from these candidates. (c) **One-Stage**. The one-stage methods like STVGBert [29] and a very recent concurrent work TubeDETR [40] tackle the STVG task through a unified architecture that directly grounds the final spatio-temporal object tube from the given video-sentence pair.

Table 1: Performance comparisons of the state-of-the-art on the VidSTG test set (%).

| Methods | Declarative Sentences | | | | Interrogative Sentences | | | |
|---|---|---|---|---|---|---|---|---|
| | m_tIoU | m_vIoU | vIoU@0.3 | vIoU@0.5 | m_tIoU | m_vIoU | vIoU@0.3 | vIoU@0.5 |
| *Factorized:* | | | | | | | | |
| GroundeR [28]+TALL [7] | | 9.78 | 11.04 | 4.09 | | 9.32 | 11.39 | 3.24 |
| STPR [38]+TALL [7] | 34.63 | 10.40 | 12.38 | 4.27 | 33.73 | 9.98 | 11.74 | 4.36 |
| WSSTG [4]+TALL [7] | | 11.36 | 14.63 | 5.91 | | 10.65 | 13.90 | 5.32 |
| GroundeR [28]+L-Net [3] | | 11.89 | 15.32 | 5.45 | | 11.05 | 14.28 | 5.11 |
| STPR [38]+L-Net [3] | 40.86 | 12.93 | 16.27 | 5.68 | 39.79 | 11.94 | 14.73 | 5.27 |
| WSSTG [4]+L-Net [3] | | 14.45 | 18.00 | 7.89 | | 13.36 | 17.39 | 7.06 |
| *Two-Stage:* | | | | | | | | |
| STGRN [48] | 48.47 | 19.75 | 25.77 | 14.60 | 46.98 | 18.32 | 21.10 | 12.83 |
| STGVT [33] | - | 21.62 | 29.80 | 18.94 | - | - | - | - |
| OMRN [47] | 50.73 | 23.11 | 32.61 | 16.42 | 49.19 | 20.63 | 28.35 | 14.11 |
| *One-Stage:* | | | | | | | | |
| STVGBert [29] | - | 23.97 | 30.91 | 18.39 | - | 22.51 | 25.97 | 15.95 |
| TubeDETR [40] | 48.10 | 30.40 | 42.50 | 28.20 | 46.90 | 25.70 | 35.70 | 23.20 |
| STCAT (Ours) | **50.82** | **33.14** | **46.20** | **32.58** | **49.67** | **28.22** | **39.24** | **26.63** |

Table 2: Performance comparisons of the state-of-the-art on the HC-STVG test set (%).

| Methods | m_tIoU | m_vIoU | vIoU@0.3 | vIoU@0.5 |
|---|---|---|---|---|
| *Two-Stage:* | | | | |
| STGVT [33] | - | 18.15 | 26.81 | 9.48 |
| *One-Stage:* | | | | |
| STVGBert [29] | - | 20.42 | 29.37 | 11.31 |
| TubeDETR [40] | 43.70 | 32.40 | 49.80 | 23.50 |
| STCAT (Ours) | **49.44** | **35.09** | **57.67** | **30.09** |

The complete performance comparisons between our model and the aforementioned approaches are presented in Tables 1 and 2 for two video benchmarks respectively. Overall, our STCAT achieves the state-of-the-art grounding performance in terms of all evaluation metrics (**m_vIoU**, **m_tIoU** and **vIoU@R**) for both two complex benchmarks, demonstrating the superiority of the proposed model. From the shown results, we can further obtain the following observations. 1) Separating the STVG task into individual spatial and temporal grounding problems achieved unsatisfactory performance in comparison to tacking them simultaneously (worse performance than both one-stage and two-stage). 2) Compared with conventional two-stage methods, the removal of object detectors brings a significant performance boost. It is mainly because that one-stage methods are end-to-end trainable and break through the restriction of the domain gap imposed by the pre-trained detectors. 3) Our proposed method outperforms all existing one-stage methods on two datasets. Although the very recent method [40] also applies the DETR architecture to STVG task, it only performs cross-modal interaction at the frame level in the encoder and implicitly models the temporal correlation through the self-attention layer in the origin DETR architecture. In contrast, our framework considers the global video context and maintains a template mechanism as queries for explicitly modeling consistency. All these novel designs contribute to the empirical superiority to [40] by large margins. Especially for the HC-STVG benchmark (*e.g.*, 30.09% v.s. 23.50% in terms of **vIoU@0.5**), since it contains a longer query sentence to describe complicated scene change within movies, delivering a higher demand for modeling subtle spatio-temporal clues.

## 4.4 Ablation Study

In this section, we conduct some ablations on the VidSTG benchmark to further investigate the contributions and design choices of the components in the proposed framework.

**Effect of the template mechanism.** To demonstrate the validation of the proposed global and local template mechanism, we design three variants of our models : 1) Remove the global template in decoder and initialize the content query with a zero vector, while remain the local template for generating position query. 2) Remove the local template in decoder and leverage a learnable position query while remain the global template as content query. 3) Remove both two types of template. The quantitative ablation results are shown in Table 3a. One can observe that cancelling either local or global template mechanism will weaken the grounding performance. Besides, the local template contributes more to the whole performance (+1.64% m_vIoU v.s. +0.95% m_vIoU). This is basically

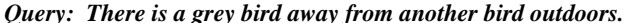

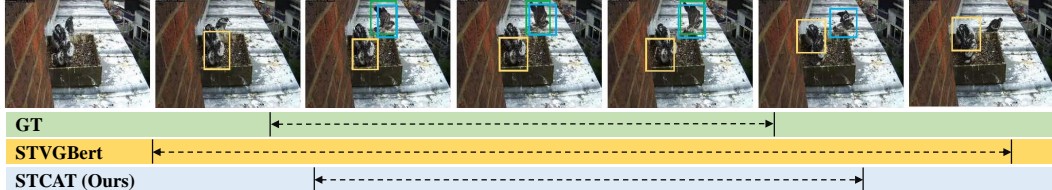

*Query: What does the adult in white clothes hug?*

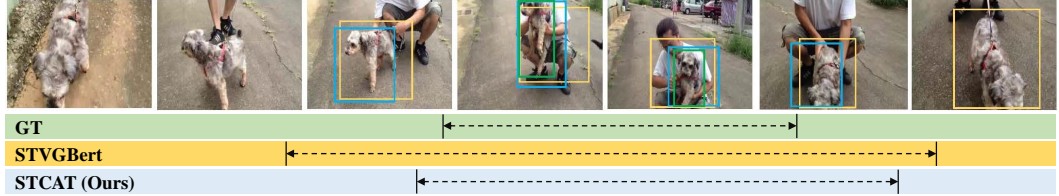

Figure 4: Some illustration examples of the spatio-temporal video grounding predictions produced by the STVGBert [29] (yellow) and our model (blue), compared with annotated ground truth (green) on VidSTG benchmark.

intuitive since the local template is specified by frame characteristic. In summary, both two types of template are benefitial for the STVG task.

| Local | Global | m_vIoU | vIoU@0.3 | vIoU@0.5 |
|-------|--------|--------|----------|----------|
|       |        | 29.59  | 41.92    | 27.21    |
| ✓     |        | 31.23  | 44.90    | 29.21    |
|       | ✓      | 30.54  | 41.52    | 28.76    |
| ✓     | ✓      | **33.14** | **46.20** | **32.58** |

(a) Ablations for the local and global templates.

| Setting | m_vIoU | vIoU@0.3 | vIoU@0.5 |
|---------|--------|----------|----------|
| w/o Temp | 31.12 | 44.36 | 29.79 |
| w/ Temp | **33.14** | **46.20** | **32.58** |

(b) Ablations for the temporal interaction layer.

Table 3: The ablation studies on VidSTG benchmark (declarative sentences) for demonstrating the effectiveness of different components in our proposed STCAT.

**Effect of the temporal interaction layer.** The temporal interaction layer in our cross-modal encoder is obliged to model the global context across the whole video, which is crucial for conducting a consistent feature alignment for encoder and maintaining a video-level objective for the decoder. We thus explore the function of temporal interaction layer via eliminating it from each encoder block (denoted as w/o temp). For the variant "w/o temp", the global embedding $p_g$ is just the average pooling of all local embedding $p_l$. The detailed ablation results are shown in Table 3b. From the presented comparison, we can observe a distinct performance drop without this layer, which further validates the effectiveness of our proposed temporal interaction layer.

## 4.5 Qualitative Analysis

In this section, we illustrate some examples in figure 4 to qualitatively compare our predictions with the one-stage method STVGBert [29] on VidSTG dataset. As shown in Figure 4, our model can predict more meaningful and accurate spatio-temporal tubes than STVGBert. Specially, for the first query, the ground truth tube should contain the top-right bird in the given video specified by "away from". However, due to the lack of long-range context modeling, the STVGBert retrieved unmatched results. In contrast, our STCAT attended the whole video content and thus can precisely localize the desired instance. More visualization examples are provided in the supplementary material.

## 5 Conclusion

In this paper, we develop an effective one-stage consistency-aware transformer network (STCAT) for tackling the spatio-temporal video grounding task. To mitigate the feature alignment and prediction

inconsistency in previous approaches, our proposed STCAT introduces a novel template mechanism to modulate the grounding procedure, which serves as a global target that guides and correlates the predictions among each individual frame to maintain temporal consistency. In addition, The extensive experimental results further demonstrate the superiority of the proposed network on the STVG task.

Acknowledgement: The research is supported by Science and Technology Innovation 2030 - New Generation Artificial Intelligence (2020AAA0104401), Beijing Natural Science Foundation (Z190001), and Peng Cheng Laboratory Key Research Project No.PCL2021A07.

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
