# Supplementary Material for "Embracing Consistency: A One-Stage Approach for Spatio-Temporal Video Grounding"

**Yang Jin[1], Yongzhi Li[2], Zehuan Yuan[2], Yadong Mu[1,3]***

[1]Peking University    [2]ByteDance    [3]Peng Cheng Laboratory

jiny@stu.pku.edu.cn, liyongzhi.ailab@bytedance.com,
yuanzehuan@bytedance.com, myd@pku.edu.cn

In the supplemental material, we firstly present a more concrete exposition for our query-guided decoding in Section 1. Then, the additional implementation details are provided in Section 2. Next, Section 3 presents more ablation study results with respect to model designs and hyper-parameter settings. To give a more intuitive demonstration, in Section 4, we show more visualization of grounding results and the attention weights in the decoder. Finally, more analysis for the comparison of results and model limitations are provided in Section 5.

## 1 More Details for Query-Guided Decoding

The detailed computation pipeline of the proposed query-guided decoding is shown in Figure 1. Given the yielded $\{q_c^t, q_p^t\}_{t=1}^{T}$ from the template generator, we formulate them as the object queries [1] to guide the overall decoding procedure inspired by recent query-based detectors [1, 4, 7]. We mainly introduce the architecture of bounding-box decoder (similar to temporal decoder). As illustrated in Figure 1, the decoder consists of $M$ stacked decoder blocks. Each block has a self-attention layer for modeling the temporal interactions across the entire video and a cross-attention layer for probing the encoded multi-modal feature within the corresponding frame.

In detail, we denote $\{Q_t\}_{t=1}^{T}$ as the object query for each individual frame. $Q_t = [C_t; P_t]$ has a content query $C_t$ and positional query $P_t$. At each decoder block, $C_t, P_t$ are firstly generated from $q_c^t, q_p^t$ based on:

$$C_t = q_c^t, \quad P_t = \text{Linear}(\text{PE}(q_p^t)), \tag{1}$$

where PE means the sinusoidal position encoding to $q_p^t = (x_t, y_t, w_t, h_t)$. Then, the content query $C_t$ is enriched by the self-attention layer and cross-attention layer, while the positional query $P_t$ serves as the position encoding like DETR [1]. In addition, following [7, 2], the positional term $q_p^t$ for generating $P_t$ is also updated in a layer-by-layer fashion via a shared prediction head. Finally, the refined $C_t$ and $P_t$ are fed into a prediction head to output the retrieved object tube. Next, the details of each individual layers are further elaborated.

**Self-Attention Layer.**   In this sub-layer, the content query $C_t$ belongs to each frame $v_t$ attends to all other frames to aggregate the global temporal context across the whole video. To make the interaction sensitive to the original temporal positions of different $C_t$, we also add a sinusoidal time encoding to each content query $C_t$. Through the self-attention layer, the grounded objects among all frames are further associated to maintain consistency.

**Cross-Attention Layer.**   In line with DETR [1], the cross-attention layer aims to aggregate the encoded cross-modal representation for enriching the content query. It is shared among all $T$ frames. Specially, given the encoded cross-modal representation $F_{vl}^t \in \mathcal{R}^{(N_v + N_s) \times C}$ with respect to frame

---

*Corresponding Author.

36th Conference on Neural Information Processing Systems (NeurIPS 2022).

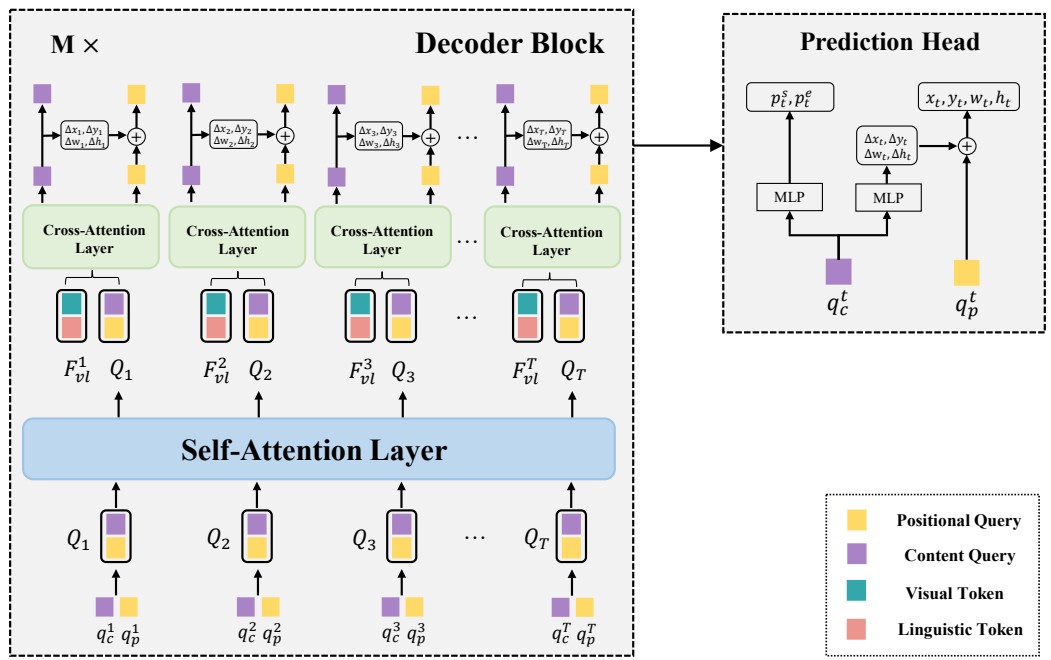

Figure 1: The Architecture of the proposed query-guided decoder and prediction head.

$v_t$, the content query $C_t$ cross-attends $F_{vl}^t$ with $P_t$ as the position encoding, where $P_t$ serves as a positional anchor to guide the decoder focus the region that contains target object most possibly. After cross-attention layer, we leverage a shared prediction head upon the content query $C_t$ to yield a relative position $(\Delta x_t, \Delta y_t, \Delta w_t, \Delta h_t)$ for updating $\boldsymbol{q_p^t} = (x_t, y_t, w_t, h_t)$, as in Figure 1.

**Prediction Head.** Through the above query-guided decoding, the final object tube is generated through a prediction head with refined content query $C_t$ as the input, which has a bounding-box branch to predict a 4-dimensional coordinate offset $(\Delta x_t, \Delta y_t, \Delta w_t, \Delta h_t)$ (shared among each decoder block as aforementioned) and a temporal branch to predict the the start probability $p_t^s$ and end probability $p_t^e$ for each frame. Each prediction branch is fulfilled by a 3-layer MLP. The final bounding box $\hat{b}_t = (x_t, y_t, w_t, h_t)$ in frame $v_t$ is obtained by adding the predicted offset and $\boldsymbol{q_p^t}$.

## 2 Additional Implementation Details

The proposed model is trained on 32 Nvidia A100 GPUs with 1 video per GPU. Besides, an AdamW optimizer [3] is used for training with weight decay $10^{-4}$. The learning rate for visual and linguistic encoders is set to $10^{-5}$, and $10^{-4}$ for the rest modules. For both VidSTG and HCSTVG benchmark, the input video is uniformly down-sampled to 64 frames for computation efficiency. We train our networks for 10 epochs on VidSTG [9] and drop the learning rate by 0.1 after 8 epochs, which consumes about 0.5 day. As for HC-STVG [6], it takes 90 epochs for training, and also a learning rate drop after 50 epochs. The whole optimization procedure costs around 6 hours. The input video resolution is set 448 under the best configuration.

## 3 More Ablation Studies

In this section, more ablation experiments are conducted to explore the influence of different frame resolutions on grounding performance. Besides, we also provide the ablation results for demonstrating the effect of our template mechanism and temporal interaction layer on the HC-STVG benchmark.

**Effect of Input Resolution.** In STVG task, the resolution of input video is a crucial influence factor for grounding performance. We conduct extensive experiments to analyze the impact of resolution

on VidSTG and HC-STVG benchmarks. The detailed results are shown in Table 1 and Table 2. When increasing the input frame resolution, the performance on VidSTG dataset will be slightly promoted accordingly. It is intuitive since a large spatial scale brings more fine-grained visual clues for multi-modal reasoning. Despite improvement, performance gains obtained by further resolution increase are reduced due to the limitation of original video resolution. As for HC-STVG dataset, the grounding accuracy seems to be less sensitive to different frame resolutions. In addition, a larger resolution even deteriorates the experimental performance (*e.g.*, 34.12% at 480 v.s. 35.09% at 448 in terms of m_vIoU on HC-STVG). Therefore, we set the input frame resolution as 448 for a trade-off between accuracy and efficiency.

**Effect of the template mechanism.** Next, we explore the effect of proposed template mechanism for HC-STVG dataset. Similar to the settings on VidSTG (See main text), three variants of our model are designed to conduct this ablation and the results are shown in Table 3a. Without the local and global template, the performance degenerates dramatically on HC-STVG benchmark (35.09% v.s. 32.12% at m_vIoU), which indicates the critical role of generated template for guiding object tube decoding and correlating the predictions among all video frames.

**Effect of the temporal interaction layer.** Finally, we provide the detailed ablation results of the temporal interaction layer for HC-STVG benchmark in Table 3b. Compared with removing the temporal interaction layer in encoder, one can observe a clear performance improvement when leveraging such a crucial module. It is worth noting that the benefit of this layer is more remarkable on HC-STVG than VidSTG (See Table 3b in the main text), since it contains more complicated query sentences describing state changes over time.

Table 1: Performance comparisons of different input resolutions on the VidSTG test set (%).

| Resolution | GPU Mem | Declarative Sentences | | | | Interrogative Sentences | | | |
|---|---|---|---|---|---|---|---|---|---|
| | | m_tIoU | m_vIoU | vIoU@0.3 | vIoU@0.5 | m_tIoU | m_vIoU | vIoU@0.3 | vIoU@0.5 |
| 320 | 19.3 GB | 49.82 | 31.85 | 45.42 | 30.61 | 48.75 | 26.99 | 37.78 | 24.77 |
| 352 | 22.3 GB | 50.18 | 32.49 | 46.10 | 31.45 | 48.88 | 27.47 | 38.20 | 25.36 |
| 384 | 26.0 GB | 49.84 | 32.16 | 45.57 | 31.63 | 48.87 | 27.55 | 38.43 | 25.54 |
| 416 | 29.1 GB | 50.57 | 32.94 | 46.07 | 32.32 | 49.23 | 27.87 | 38.89 | 26.07 |
| 448 | 30.4 GB | **50.82** | **33.14** | **46.20** | **32.58** | **49.67** | **28.22** | **39.24** | **26.63** |
| 480 | 33.4 GB | 50.52 | 32.86 | 46.01 | 32.41 | 49.52 | 28.05 | 38.49 | 26.00 |

Table 2: Performance comparisons of different input resolutions on the HC-STVG test set (%).

| Resolution | GPU Mem | m_tIoU | m_vIoU | vIoU@0.3 | vIoU@0.5 |
|---|---|---|---|---|---|
| 320 | 23.7 GB | 48.24 | 33.81 | 55.09 | 29.48 |
| 352 | 25.2 GB | 48.98 | 34.75 | 55.95 | 30.34 |
| 384 | 28.0 GB | 48.09 | 34.06 | 54.48 | 29.31 |
| 416 | 30.7 GB | 48.85 | 34.93 | 56.64 | **31.03** |
| 448 | 32.1 GB | **49.44** | **35.09** | **57.67** | 30.09 |
| 480 | 34.8 GB | 48.80 | 34.12 | 55.78 | 29.57 |

| Local | Global | m_vIoU | vIoU@0.3 | vIoU@0.5 |
|---|---|---|---|---|
| | | 32.12 | 51.46 | 23.28 |
| ✓ | | 34.05 | 55.93 | 26.83 |
| | ✓ | 33.16 | 54.28 | 22.37 |
| ✓ | ✓ | **35.09** | **57.67** | **30.09** |

(a) Ablations for the local and global templates.

| Setting | m_vIoU | vIoU@0.3 | vIoU@0.5 |
|---|---|---|---|
| w/o Temp | 33.64 | 54.83 | 26.12 |
| w/ Temp | **35.09** | **57.67** | **30.09** |

(b) Ablations for the temporal interaction layer.

Table 3: The ablation studies on HC-STVG benchmark (%) for demonstrating the effectiveness of different components in our proposed STCAT.

## 4 More Visualization Results

### 4.1 Visualization for Attention Weights

In order to gain more insights of proposed template mechanism, we visualize the cross-attention weights produced by decoder. These attention maps clearly show the difference of mostly attended

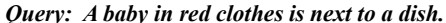

*Query: A baby in red clothes is next to a dish.*

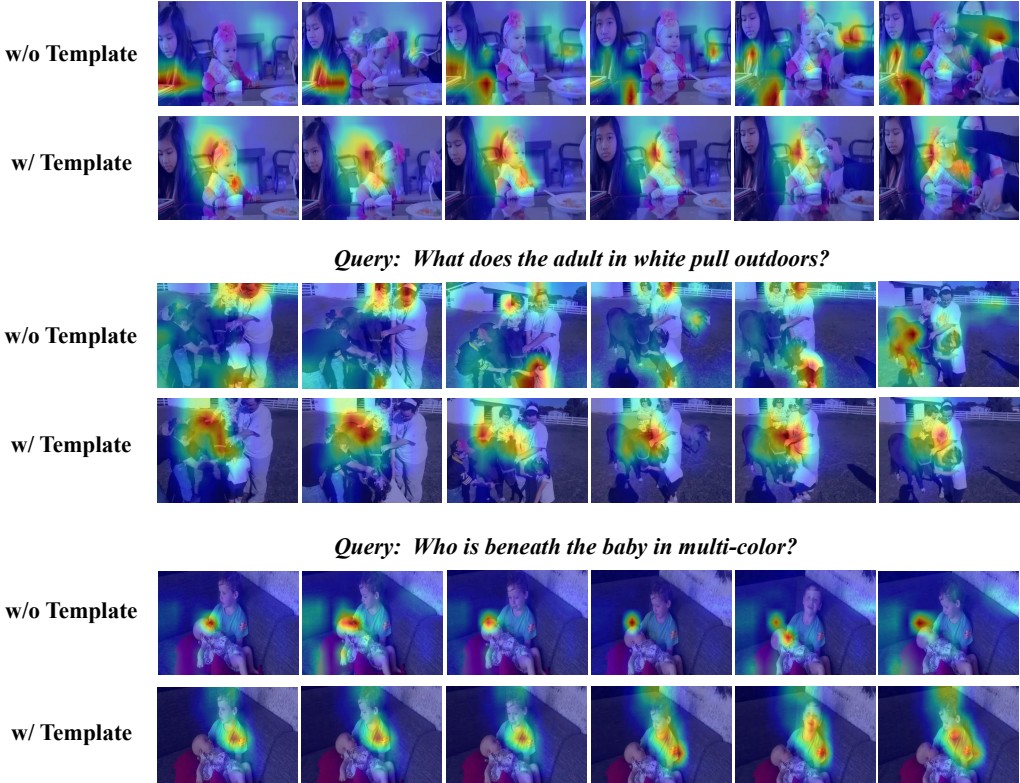

Figure 2: The visualization of cross-attention weights in the decoder. For each case, the top row indicates attention weights without proposed template mechanism (denoted as w/o Template) and the bottom row indicates model with the generated template as object query (denoted as w/ Template).

regions when leveraging the generated template as object queries for decoding (dubbed as w/ Template) or not (dubbed as w/o Template). As illustrated in Figure 2, we can clearly find that the proposed template mechanism is quite crucial for settling STVG task. After adopting the generated template as object queries, the attention is highly concentrated on the target interest regions, which successfully associate and restrict the grounding results among all video frames to maintain consistency. For example, given the sentence "What does the adult in white pull outdoors?", the attention of our model (w/ Template) is particularly focused on the "horse" that is being pulled out. However, the model simply with learnable queries (w/o Template) is distracted by the unrelated person and horse, leading to unsatisfactory grounding results.

## 4.2 Visualization for Grounding Results

To further qualitatively validate the effectiveness of proposed approach, more illustration examples of grounding results in comparison with the recent one-stage method STVGBert [5] on VidSTG dataset are provided in Figure 3. From the shown visualizations, it can be evidently observed that our model can produce more accurate predictions corresponding to the given sentence queries for STVG task. In detail, for the first example, the desired object tube should contain "the baby in red", while STVGBert failed to localize the target and only attended wrong "adult woman in black". In contrast, our model is able to better reason the sentence semantics and subtle clues in video, thus successfully grounding the desired "baby in red". The same comparison can also be found in the 4-th example. Due to insufficient global context modeling and inconsistent feature alignment, STVGBert can not accurately retrieve the target "child" from several ambiguous candidates. As for temporal localization, our STCAT also generated more precise segment boundaries than STVGBert, demonstrating the superiority of proposed model in dealing with complicated temporal-text interactions.

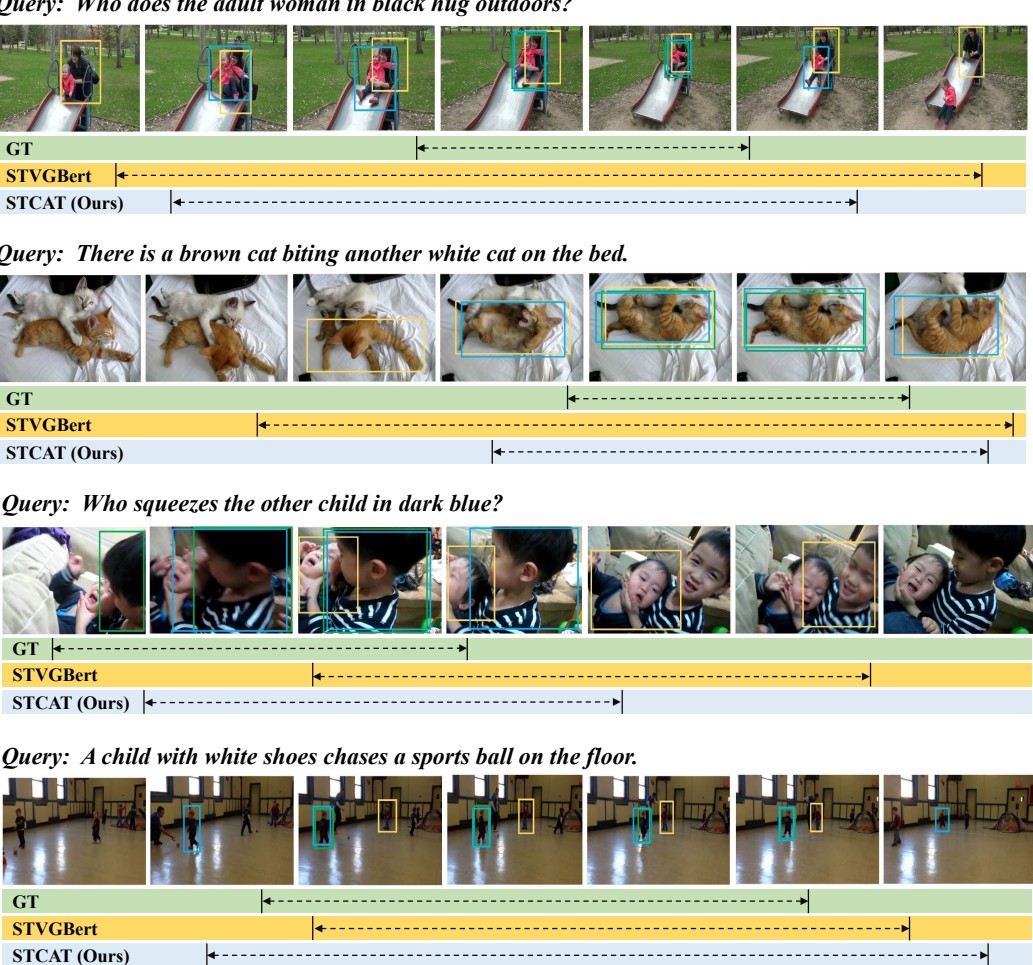

Figure 3: Some illustration examples of the spatio-temporal grounding results produced by STVG-Bert [5] (yellow) and our model (blue), compared with ground truth (green) on VidSTG benchmark.

## 5  Additional Analysis

### 5.1  Comparison with Contemporaneous Work

More recently, a concurrent work in [8] termed as TubeDETR also applies the DETR architecture to settle STVG task [2]. However, they still suffer from feature alignment and prediction inconsistency deficiencies like all existing approaches. Next, we will briefly discuss the differences. (1) During the multi-modal feature encoding, they only consider the multi-modal interaction within every single frame while not considering the global video context, which inevitably brings feature alignment inconsistency (See Section 1 in our main text). In contrast, the proposed STCAT introduce a local learnable token to attend the local context within the individual frame and a global one to integrate the global video context during encoding. Coupling these tokens with the spatial and temporal interaction layer in encoder, our framework is able to perform a more consistent feature alignment between two modalities. (2) The TubeDETR only adopts the original learnable query in DETR to decode the target object tube. Even for the same video, given different query sentences, the query used for decoding still keeps invariable in TubeDETR. This implementation is unsuitable for settling the STVG, since this task requires models to retrieve one text-specific instance. Simply leveraging the text-agnostic object query to decode the desired object for each frame in parallel will result in an inconsistent

---

[2]The results in original paper [8] is incorrect due to the wrong evaluation code when we write this paper. We report the correct performance updated by the author in this https://github.com/antoyang/TubeDETR/issues/3.

prediction. Our STCAT introduces a novel multi-modal template as the global objective to constrict and associate the prediction at each frame and thus produce more consistent grounding results. (3) The proposed framework outperforms TubeDETR on two large STVG benchmarks by a large margin (See Table 1 and 2 in main text). Especially for the HC-STVG benchmark (*e.g.*, 48.74% v.s. 43.70% at m_tIoU, 31.98% v.s. 23.50% at vIoU@0.5), since it contains longer query sentence to describe a complicated scene change within movies, delivering a higher demand for modeling long-term video context and subtle multi-modal reasoning.

## 5.2 Limitations

The limitations in this paper lie in the localization precision of temporal boundary, which exists in all STVG approaches. It is mainly because timing is inherently ambiguous and often difficult to specify a particular start and end frame. In detail, for the first example in Figure 3, it is quite hard to localize the ground-truth segment since the boundary for "hug" is very ambiguous and subjectively determined by the dataset annotators.