# OpenReview forum: "Embracing Consistency: A One-Stage Approach for Spatio-Temporal Video Grounding"
_NeurIPS.cc/2022/Conference — NeurIPS 2022 Accept_

### Official Review · Reviewer_5BXu · 2022-07-11

**Rating:** 6
**Confidence:** 4
**Soundness:** 3 good
**Presentation:** 3 good
**Contribution:** 2 fair

**Summary:**

This paper introduces a new algorithm, STCAT, to solve the Spatio-Temporal video grounding problem. The new approach is based on the DETR implementation and achieves state-of-the-art performance. By considering the video's global information, it is expected to solve the problem of Prediction Inconsistency, i.e., prediction is performed only at the frame-level, ignoring the consistency between frames.

**Questions:**

1. I find that there is no explicit loss to enforce the feature consistency, I wonder how your designed module can make the feature “consistency”. I think more visualization and analyze deserve to be done for this part.
2. In the ablation study, the effect of the temporal interaction layer section, what is the global embedding pg if the model is trained without the temporal interaction layer.

**Strengths And Weaknesses:**

Strengths:
1. The newly proposed method achieves state-of-the-art performance.
2. The article is very clearly written and easy to understand.
Weaknesses:
1. The ablation study is insufficient, and we do not know the effect of the prediction head.
2. The method is constructed on the top of DETR, so the paper contributes less to the network structure.

---

> ### Author Response · Authors · 2022-08-02
> **Response to reviewer 5BXu**
>
> We thank the reviewer for appreciating our work and providing valuable feedback. Below are our responses to the raised concerns.
>
> *Q-1: I find that there is no explicit loss to enforce the feature consistency, I wonder how your designed module can make the feature “consistency”. I think more visualization and analysis deserve to be done for this part.*
> * The "feature consistency" in this work refers to the cross-modal feature alignment consistency, which aims to perform multi-modal fusion between the whole video content and the querying sentence. For the STVG task, cross-modal feature alignment is essential. The model can correctly retrieve the target object only under sufficient video-textual perception. However, existing methods only consider the multi-modal fusion with local short-term frame context and ignore global video content, which brings the inconsistency issue. Our model achieves better consistent feature alignment through the stack of spatial and temporal interaction layers in the encoder. These two kinds of layers cooperate with each other to fuse the local and global visual clues with textual semantics.
> * We are not fully confident about the implication of "feature consistency" when mentioned by the reviewer. We guess that it refers to the consistency among the visual feature of the queried object from different frames. In our current implementation, this consistency is implicitly constrained by the proposed template mechanism, where each frame shares the same content term $q_c$ in generated template. The shared $q_c$ represents the semantics of queried object. It serves as the global objective in our decoder to ground the common target for all video frames. The visualization results in Section 4.1 of the supplemental material further show that the attention is highly concentrated on the target object after adopting the generated template. We fully agree with the Reviewer that adding some loss function (e.g., the similarity of object feature between frames should be higher) to explicitly enforce the feature consistency may be beneficial. We will include more analysis in the next revision as suggested.
>
> *Q-2: In the ablation study, the effect of the temporal interaction layer section, what is the global embedding pg if the model is trained without the temporal interaction layer.*
> * We regret for the ambiguous description of experimental setting. Actually, if the model is trained without the temporal interaction layer, the global embedding $p_g$ is just the average pooling of all local embedding $p_l$. We will clarify it in the revision.
>
> *Q-3: The ablation study is insufficient, and we do not know the effect of the prediction head.*
> * Thanks for pointing out the insufficiency of our ablation study. The prediction head consists of a box branch to regress the 4-dimensional coordinates for the object bounding-box and a temporal branch to predict the start and end probability for each frame. Each branch is fulfilled by the MLP and is indispensable for generating the retrieved tube. We conduct another ablation study on HC-STVG dataset to explore the impact of MLP layers on model performance.
>
> Layer Number | Bounding-box Head | Temporal Head
> :---: | :---: | :---:
> 1 | 32.87  | 33.13
> 2 | 34.92 | 35.35
> 3 | **35.35** | **35.39**
> 4 | 35.14 | 35.13
>
> * From the shown results, we can observe that 3-layer MLP is the best configuration for both spatial and temporal prediction heads. Stacking more MLP layers is useless for boosting model performance. We will include these ablation results in the next revision.
>
> *Q-4: The method is constructed on the top of DETR, so the paper contributes less to the network structure.*
> * Although our model employs the same encoder-decoder paradigm like DETR, the architecture we propose is specifically tailored for more consistent video grounding. This is reflected in the design of both encoder and decoder. The encoder introduces local and global learnable tokens to attend to the local context and global video semantics simultaneously. Through the stack of spatial and temporal interaction layers, it provides sufficient cross-modal feature alignment. The decoder integrates the video-level multi-modal template yielded from the proposed template generator module as object queries to guide the tube generation. Profiting from these elaborate designs, our STCAT outperforms all the previous approaches on STVG task.

---

### Official Review · Reviewer_wsdA · 2022-07-12

**Rating:** 6
**Confidence:** 3
**Soundness:** 3 good
**Presentation:** 3 good
**Contribution:** 2 fair

**Summary:**

The paper presents an end-to-end one-stage Transformer-based framework for spatio-temporal video grounding (STVG). In particular, the authors introduced a novel multi-modal template to alleviate the alignment inconsistency and prediction inconsistency issues in the task above. The model followed an encoder-decoder design paradigm without any pre-trained object detectors, which reached resonable performance on both VidSTG and HC-STVG datasets.

**Questions:**

Please see the *Strengths And Weaknesses* section.

**Limitations:**

Yes,  the authors adequately addressed the limitations.

**Strengths And Weaknesses:**

**Strengths**
- The main contribution is the semantic-consistency flow pattern between the encoder and decoder for the proposed Spatio-Temporal Consistency-Aware Transformer (STCAT) architecture, which mainly consists of an encoder, a template generator, and a corresponding decoder with couples of prediction heads.
- The encoder utilized two specifically-designed interaction layers (spatial interaction layer and temporal interaction layer) to enhance the visual-linguistic feature fusion and the semantic consistency intra- and inter-frames. The template generator produced two terms for both global query representation and frame-level prediction constraint, which benefitted the decoder for spatial-temporal grounding.
- STCAT sets new state-of-the-art performance on the two aforementioned datasets under most evaluation metrics.
- The paper is well-written and easy to follow. The ablation study is also reasonable.

**Weaknesses**
- I am confused about the selection of ResNet instead of ViT as the visual feature extractor. ViT can naturally provide well-organized and adequately-interacted tokens for each frame, which can promote the feature quality for encoder layers. So why not use a transformer-fashion backbone for visual features?
- The naming of the “Spatial Interaction Layer” seems a bit imprecise to me. It seems this module is mainly designed for visual and linguistic semantic fusion instead of boosting spatial information interaction in each frame. If the spatial information of visual tokens needs to be enriched before inter-frame interaction, I sincerely suggest you consider ViT-fashion models.
- Detailed information about the input for the cross-attention layer in each decoder block might need to be supplied.
- The usage of certain loss functions also seems unreasonable. What’s the aim of combining $L_1$loss with GIoU loss for bbox prediction? If the $L_1$loss is complementary to the GIoU loss for eliminating certain special cases when supervising via IoU,  the CIoU or DIoU might be a better metric than the combination, which also avoids tolerating the slow converge speed of GIoU.
- The comparison between STCAT with other one-stage attention-based models seems insufficient. For example, what’s the throughput and parameter amount for STCAT to reach better performance than TubeDETR and STVGBert?
- Could the authors provide an efficiency analysis such as FLOPs, latency and running time on the proposed method and the baselines to provide prove the effectiveness of the proposed method.

---

> ### Author Response · Authors · 2022-08-02
> **Response to reviewer wsdA**
>
> We would appreciate the encouraging review and insightful questions. Here are our responses to your main concerns.
>
> *Q-1: I am confused about the selection of ResNet instead of ViT as the visual feature extractor. ViT can naturally provide well-organized and adequately-interacted tokens for each frame, which can promote the feature quality for encoder layers. So why not use a transformer-fashion backbone for visual features?*
> * We fully agree with the reviewer that ViT can naturally provide adequately-interacted tokens and thus can serve as a better visual feature extractor. The reason we choose ResNet is for fair performance comparison with others since all the existing STVG approaches leverage it as their visual backbone. We will include the ViT-based architecture as model variants and compare the performance with ResNet in future revision as suggested.
>
> *Q-2: The naming of the "Spatial Interaction Layer" seems a bit imprecise to me. It seems this module is mainly designed for visual and linguistic semantic fusion instead of boosting spatial information interaction in each frame. I sincerely suggest you consider ViT-fashion models.*
> * Thanks for pointing out the name ambiguity here. We dubbed this layer as "Spatial Interaction Layer" to emphasize the multi-modal fusion within local frame level and to contrast with the utility of temporal interaction layer. Maybe the name "Cross-Modal Interaction Layer" is much preciser. Meanwhile, we will revise our manuscript to integrate some ViT-fashion models in the next version following your valuable suggestion.
>
> *Q-3: Detailed information about the input for the cross-attention layer in each decoder block might need to be supplied.*
> * The detailed input for the cross-attention layer is provided in Section 1 of the supplemental material. It takes the encoded cross-modal representation $F_{vl}^t \in \mathcal{R}^{(N_v + N_s) \times C}$ with respect to frame $v_t$, the content query $C_t$ and the position query $P_t$ as input, where $C_t$ cross-attends $F_{vl}^t$ with $P_t$ as the position encoding. Thanks for pointing this and we will include this critical information to the main text in the revision.
>
> *Q-4: The usage of certain loss functions also seems unreasonable. What’s the aim of combining $L_1$ loss with GIoU loss for bbox prediction? If the $L_1$ loss is complementary to the GIoU loss for eliminating certain special cases when supervising via IoU, the CIoU or DIoU might be a better metric than the combination, which also avoids tolerating the slow converge speed of GIoU.*
> * Following the previous common practice in DETR [1] and many other transformer-based detection approaches [2, 3], we leverage both Smooth $L_1$ and GIoU loss as the bounding box supervision. The combination of Smooth $L_1$ and GIoU is to comprehensively consider the absolute coordinate alignment and box scale consistency.
> * We fully agree with the reviewer that CIoU and DIoU may admit faster convergence and can be more effective than GIoU on the STVG task. The current utilization of GIoU is primarily for a fair comparison with existing methods. We thank for this constructive suggestion and will include new results by combing the $L_1$ and DIoU (or CIoU) losses in the revision.
>
> *Q-5: The comparison between STCAT with other one-stage attention-based models seems insufficient. For example, what’s the throughput and parameter amount for STCAT to reach better performance than TubeDETR and STVGBert?*
> * We list the parameter amount compared with TubeDETR [5] in the Table below (STVGBert [4] does not provide the parameter amount of their model). From the presented results, it can be seen that our STCAT has a comparable parameter amount with TubeDETR while achieving better grounding performance.
>
> |Method|Params(M)|GFLOPs|Epochs|Training Time|
> |:-------:|:-----:|:-----:|:-----:|:-----:|
> | TubeDETR| 142 | 16.8 | 10 |2 day |
> | STVGBert|  -  |   - |50|-|
> | STCAT(Ours) | 155 |    16.9 | 8 | 1.2 day|
>
> *Q-6: Could the authors provide an efficiency analysis such as FLOPs, latency and running time on the proposed method and the baselines to provide prove the effectiveness of the proposed method.*
> * We list the efficiency analysis of our STCAT and other one-stage approaches in the Table above. The GFLOPs metric is evaluated with text length 7 and video resolution 224 as input for all compared models. The detailed efficiency comparison shows that our STCAT requires less training epochs and running time for convergence.
>
> [1] End-to-end object detection with transformers, ECCV, 2020.
>
> [2] Deformable detr: Deformable transformers for end-to-end object detection, ICLR, 2021.
>
> [3] Dynamic detr: End-to-end object detection with dynamic attention, ICCV, 2021.
>
> [4] Stvgbert: A visual-linguistic transformer based framework for spatio-temporal video grounding, ICCV, 2021.
>
> [5] Tubedetr: Spatio-temporal video grounding with transformers, CVPR, 2022.

---

### Official Review · Reviewer_JJPQ · 2022-07-12

**Rating:** 6
**Confidence:** 4
**Soundness:** 3 good
**Presentation:** 4 excellent
**Contribution:** 3 good

**Summary:**

The authors proposed to approach the problem of spatiotemporal video grounding by a one stage end-to-end framework. The model adopts an encoder-decoder strategy and generates the video tube directly from output. Comparisons are made against other approaches to demonstrate the performance of the proposed approach.

**Questions:**

1) In terms of the model architecture contributions, if I understood correctly, the analogy is to add the decoder part similar to the one in traditional transformer on top of the encoder in ViT-based models and use it across different modalities, right? If that's the case, could you elaborate what specific modifications need to be done that can impact the performance of the model?

2) For results in Table 1, when compared with OMRN, the other metrics are clearly outperforming, but it's not the case for m_tIoU. It was mentioned that "OMRN leverages an off-the-shelf detector", so can you explain why the detector mainly helps with one specific metric, or does this indicate that STCAT and OMRN are actually performing differently on different types of test data? If so, some qualitative comparisons might be helpful for error analysis and intuitive understanding.

3) For the generated tube from the decoder, is there any kind of constraint applied so that the output is more consistent across frames, or is it simply enforced by attention? In other words, for other frame-based models, if some simple heuristic constraints can be added to output frames, would that also solve the inconsistency issue, at least to some extent?

**Limitations:**

The authors addressed the limitations in Supplementary Section 5.2.

**Strengths And Weaknesses:**

The paper is generally well written, the model architecture is intuitive and the results are reasonably well.

---

> ### Author Response · Authors · 2022-08-02
> **Response to reviewer JJPQ**
>
> We thank the reviewer for the positive and constructive comments. Below are our responses to the raised concerns.
>
> *Q-1: The model architecture contributions.*
> * Although the proposed model adopts a transformer-fashion architecture, both the encoder and decoder possess tailored configurations for tackling the spatio-temporal video grounding task. The encoder acts as a contextual information interactor across vision and language modalities. Compared to the one in ViT-based models, we introduce a local learnable token to attend the local context within the individual frame and a global token to integrate the global video context during encoding. Coupling with the spatial and temporal interaction layer, the encoder can fully exploit the global video context and thus perform a better cross-modal alignment.
>
> * The decoder has self-attention and cross-attention modules similar to traditional transformer, but the query $Q$ in these modules are designed specially for the interested task. In specific, in the proposed model, the query vector $Q_t=[C_t;P_t]$ is adaptively generated from the so-called Template Generation module, which serves as a global target to guide the attention layer. During decoding, $Q_t$ is updated layer-by-layer to obtain the final contextualized representation used for generating object tubes. More technical details about the decoder can be found in Section $1$ of the supplementary material.
> * The ablation studies in Table 3 reveal a few observations regarding the impact of model configurations on performance. For example, if the temporal interaction layer in the encoder or the generated template in the decoder were removed, the grounding performance will both deteriorate a lot (-2.66% and -1.22% on m\_vIoU respectively).
>
> *Q-2: The m_tIoU compared with OMRN.*
> * The underlying reason is not "OMRN leverages an off-the-shelf detector". We regret that such an expression made you misunderstood. The intention of this statement is to show that the proposed method is end-to-end trainable compared with OMRN. Actually, OMRN [1] additionally designs a specific module for temporal localization (different from the module for spatial localization), which leverages handcrafted sliding windows with different widths to sample a set of video segments. The temporal boundary is finally determined based on these segments. Such design only considers a finite segment width and is sensitive to hyper-parameter settings (e.g., window width) on different datasets. Therefore, compared with our model, it achieves better m\_IoU on Declarative Sentences (50.73 v.s. 50.58 ) and worse performance (49.19 v.s. 49.40) on Interrogative Sentences. In contrast, our model predicts the start and end probability for each frame and thus can localize boundaries of arbitrary length. We will make this reason analysis more convincing and include more intuitive demonstrations in the revision following your valuable suggestion.
>
> *Q-3: For the generated tube from the decoder, is there any kind of constraint applied so that the output is more consistent across frames, or is it simply enforced by attention?*
>
> * There is no heuristic constraint on the generated tube. We directly use the decoder outputs as the final grounding results. We fully agree with the reviewer that some simple heuristic constraints may be helpful to alleviate the inconsistency issue. Actually, some frame-based models like [1, 2, 3] have already adopted some heuristic post-process (e.g., the bounding box IoU between adjacent frames should be greater than a threshold) to generate the final object tube. Although such constraint can solve the inconsistency issue to some extent, it heavily relies on the compatibility between all the initial bounding boxes predicted by the model at one frame and the target query object. Besides, this heuristic constraint is typically sensitive to hyper-parameters, it is non-trivial to find an optimal IoU threshold. In contrast, the proposed STCAT introduces a multi-modal template mechanism as the global objective to mitigate the prediction inconsistency issue, which is hyper-parameter-free and can be optimized via end-to-end training. Without any post-processing strategy, our STCAT still outperforms all the existing approaches on two video benchmarks.
>
> [1] Object-aware multi-branch relation networks for spatio-temporal video grounding, IJCAI, 2021.
>
> [2] Where does it exist: Spatio-temporal video grounding for multi-form sentences, CVPR, 2020.
>
> [3] Human-centric spatio-temporal video grounding with visual transformers, IEEE Transactions on Circuits and Systems for Video Technology, 2021.

---

### Meta-Review · Area_Chair_tpv2 · 2022-08-23

**Recommendation:** Accept
**Confidence:** Certain

**Metareview:**

This paper presents a novel end-to-end framework for Spatio-Temporal video grounding where the feature alignment and prediction inconsistency can be jointly addressed. All of our reviewers believe this paper is well-presented with a novel idea and SOTA results. Overall, I would like to recommend this paper to be accepted with a poster presentation.

**Award:**

No

---

### Decision · Program_Chairs · 2022-09-14

Accept